# Determination of Antioxidant, Cytotoxicity, and Acetylcholinesterase Inhibitory Activities of Alkaloids Isolated from *Sophora flavescens* Ait. Grown in Dak Nong, Vietnam

**DOI:** 10.3390/ph15111384

**Published:** 2022-11-10

**Authors:** Phan Nguyen Truong Thang, Viet-Hung Tran, Tran Anh Vu, Nguyen Ngoc Vinh, Duyen Thi My Huynh, Duy Toan Pham

**Affiliations:** 1Institute of Drug Quality Control Ho Chi Minh City, Ho Chi Minh City 700000, Vietnam; 2Faculty of Pharmacy, Hong Bang International University, Ho Chi Minh City 700000, Vietnam; 3Faculty of Pharmacy, Nguyen Tat Thanh University, Ho Chi Minh City 700000, Vietnam; 4Department of Pharmaceutical and Pharmaceutical Technology, Faculty of Pharmacy, Can Tho University of Medicine and Pharmacy, Can Tho 900000, Vietnam; 5Department of Chemistry, College of Natural Sciences, Can Tho University, Can Tho 900000, Vietnam

**Keywords:** *Sophora flavescens* Ait., antioxidant, cytotoxicity, acetylcholinesterase inhibition, Vietnam

## Abstract

Traditional/herbal medicine has gained increasing interests recently, especially in Asian countries such as Vietnam, due to its diverse therapeutic actions. In the treasure of Vietnamese medicinal plants, one of the potential herbs is the roots of *Sophora flavescens* Ait. (SF, “Kho sam” in Vietnamese). However, limited information has been reported on the Vietnamese SF compositions and their respective alkaloids’ anti-acetylcholinesterase action. Thus, this study investigated the extractions, isolations, identifications, and in-vitro antioxidant, cytotoxicity, and acetylcholinesterase inhibitory activities, of the SF root extracts and their purified alkaloid compounds. To this end, four pure compounds were successfully isolated, purity-tested by HPLC, and structurally identified by spectroscopic techniques of FTIR, MS, and NMR. These compounds, confirmed to be oxysophocarpine, oxymatrine, matrine, and sophoridine, were then determined their therapeutic actions. The SF extracts and the compounds did not possess significant antioxidant activity using the DPPH and MDA assays, and cytotoxicity action using the MTT assay on the MDA-MB-231 breast cancer cell line. On the other hand, the SF total extract yielded a moderate acetylcholinesterase inhibition effect, with an IC50 of 0.1077 ± 0.0023 mg/mL. In summary, the SF extract demonstrated potential effects as an anti-acetylcholinesterase agent and could be further researched to become a pharmaceutical product for diseases related to acetylcholine deficiency, such as dementia.

## 1. Introduction

Recently, approximately 80% of people in developing countries utilize traditional/herbal medicine (i.e., medicine that is mainly based on plants and animals) for their primary health care [1,2]. Additionally, the World Health Organization (WHO) recommends and encourages the use of traditional medicines due to their availability, potential low toxicity, and diverse therapeutic efficacy. Vietnam is a country with rich and diverse plant resources, especially therapeutic plants, which are very favorable conditions for the research and development of traditional medicines [3,4]. In the treasure of Vietnamese medicinal plants, one of the potential herbs that has gained much interest is the roots of *Sophora flavescens* Ait. (SF, commonly known as “Kho sam” in the Vietnamese language). SF has been widely used in China and Japan for a long time and has been cultivated in Vietnam since the 1990s, mostly in mountainous areas such as Dak Nong and Sapa. Ethnologically, the local people use SF root, which has a strong, bitter taste, and cold properties, in folk remedies such as Ninh Tam Vuong, which helps in tachycardia and in the treatment of gynecological infections. Clinically, SF has been scientifically proven to possess numerous therapeutic effects [5], including cytotoxicity [6], anti-inflammation [7], anti-asthma [8], anti-anaphylaxis [9], anti-microbial [10], anti-viral [11], anti-arrhythmic [12], and anti-myocardial fibrosis [13]. Commercially, SF roots have been employed in various functional products, such as Kushen food supplement of Nature’s Health, Palm Springs, CA, USA.

Among the two main compound groups in SF, the flavonoids and the alkaloids, the alkaloids have obtained increasing attention due to their outstanding effects. The SF alkaloids are classified based on their structures, mainly including four subtypes: matrin type, cytisin type, anagyrin type, and sophocarpine type. Especially those with matrin-type structures have been identified as biologically active components, possessing many pharmacological effects such as cytotoxicity, anti-inflammatory, and antioxidant activities.

Generally, plants are complex in composition, and their therapeutic activity heavily depends on their chemical constituents, which in turn correspond to the plant age, geographical location, and harvesting procedure [14,15,16]. In addition, the improper use of fertilizers and contamination by microorganisms and pesticide residues have made the medicinal plant chemical content, both in quality and quantity, significantly differ from one location to another. Therefore, although SF has been previously investigated worldwide, limited information has been reported on Vietnamese SF in terms of its alkaloid compositions and their therapeutic actions. Moreover, as far as we know, the anti-acetylcholinesterase effects of the SF alkaloid-rich extract, as well as its pure isolated alkaloids, have not yet been investigated.

Therefore, this work investigated the optimal conditions for the extraction, isolations, and purifications of the alkaloids from the SF roots. The chemical structures of the pure compounds were identified by Fourier-transform infrared spectroscopy (FTIR), mass spectrometry (MS), and nuclear magnetic resonance (NMR) techniques. These extracts and the pure compounds were then in-vitro tested for their therapeutic effects on three distinct activities of antioxidant, cytotoxicity, and acetylcholinesterase inhibition.

## 2. Results and Discussions

### 2.1. Plant Identifications

The collected SF roots possessed a long cylindrical shape, often with many branches in the lower part, with a 1.0–6.5 cm length in diameter. The external section had a yellow-gray or yellow-brown color, with longitudinal wrinkles and no accessory roots. The root structure was solid, and its cross-section had protrusions resembling air vessels (Figure 1). These characteristics were in good agreement with the literature [17]. Using the DNA sequencing technique, the SF total DNA sample gave the upper 10 kb line, and the amplification sample gave the 600 bp line (Figure 2). Moreover, compared with the gene library (code AB127037.1), the sample DNA sequencing results yielded a degree of similarity of 99%, with a nucleotide similarity ratio of 1025/1025 and an E-value of 0.0. These results indicated that the collected SF was the correct species.

### 2.2. Plant Extractions, Isolations, and Alkaloids Purifications

The yields of the SF total extract and chloroform fraction were significantly affected by the extracting solvents, namely water, water + 1% HCl, water + 1% acetic acid (AcOH), EtOH, EtOH + 1% HCl, EtOH + 1% AcOH, MeOH, MeOH + 1% HCl, and MeOH + 1% AcOH (Table 1). Generally, different types of solvents did not affect the extraction masses for both the total extract and the chloroform fraction. On the other hand, compared to the addition of AcOH, the addition of HCl in the solvents increased the extraction efficiency in the total extract but not in the chloroform fraction. Conclusively, due to its mild acid strength, AcOH was selected, and the EtOH + 1% AcOH was the optimal extraction solvent to obtain the highest yield of the chloroform fraction.

From 42.3 g of alkaloid-rich chloroform fraction, 10.4 g of yellow crystalline substances and 30.1 g of non-crystallized concentrate were obtained. Both the crystalline and non-crystallized portions were determined their compositions by thin-layer chromatography with a mobile phase of CHCl_3_-MeOH-NH_4_OH 25% (50:10:3, lower layer) (Appendix A) and high-performance liquid chromatography (HPLC, Shimadzu LC-20A machine, Phenomenex Gemini NX C18 (250 × 21.2 mm, 5 μm), UV detector at 210 nm, injection volume of 200 µL, and flow rate of 7 mL/min, for both portions) (Appendix A). The mobile phases for the crystalline portion were MeOH-trifluoroacetic acid 0.015% (6:94 *v*/*v*), and for the non-crystallized portion, was MeOH-trifluoroacetic acid 0.015% (gradient mode, 5:95 *v*/*v*-0-to-30 min, 50:50 *v*/*v*-30-to-60 min, 5:95 *v*/*v*-60-to-70 min).

Finally, from 1.562 g of the crystallization fraction, through the preparative HPLC, 116 mg of substance A1 and 626 mg of substance A2 were obtained. Similarly, from the non-crystallized fraction, 810 mg of A3 and 310 mg of A4 powders was collected. These four compounds were further identified in their structures in the next section.

### 2.3. Isolated Compound Structure Identifications

The four isolated compounds (A1–A4) were tested for their purity in HPLC prior to structure identifications. For this purity test, all four compounds showed high purity with only one characterized peak in the HPLC chromatograms (Appendix A), indicating that they are pure enough for the chemical determinations.

Then, these four compounds fully identified their structures based on the FTIR, MS, and NMR methods (Table 2 and Table 3). The compounds A1–A4 were determined as oxysophocarpine, oxymatrine, matrine, and sophoridine, correspondingly. These compound NMR data were in good agreement with the literature, which re-confirmed our findings.

### 2.4. In-Vitro Antioxidant Test

For both the DPPH and MDA antioxidant assays, both the extracts and the four pure compounds (oxysophocarpine, oxymatrine, matrine, and sophoridine) did not possess significant in-vitro antioxidant effects, with the efficacy of less than 50% at a tested concentration of as high as 1000 µg/mL (Table 4). This indicates that the SF roots might not be a potential antioxidant source in our case. Controversially, a previous study has demonstrated that both the SF methanolic extract and its dichloromethane fraction possessed DPPH radical-scavenging effects in a dose-dependent manner [21]. These differences could be attributed to the impacts of geographic variations on the physicochemical properties of SF.

### 2.5. In-Vitro Cytotoxicity Test

Compared to the positive control doxorubicin, with possessed an IC50 of 1.12 ± 0.03 µM, both the extracts and the four pure compounds (oxysophocarpine, oxymatrine, matrine, and sophoridine) showed much higher IC50 of >100 µg/mL on the MDA-MB-231 breast cancer cell line. Specifically, at the highest investigated the concentration of 100 µg/mL, the inhibitory percentages of the SF total extract, the SF chloroform fraction, and the pure compound oxysophocarpine, oxymatrine, matrine, and sophoridine were 8.25 ± 0.51%, 18.32 ± 0.97%, 20.81 ± 1.02%, 19.45 ± 0.77%, 21.30 ± 1.11%, and 20.29 ± 0.84%, respectively (Figure 3). Therefore, the SF roots demonstrated limited effects on the breast cancer cell lines. Interestingly, although SF and its alkaloids have been proven to possess high cytotoxicity in several cancer cell lines through various mechanisms, they showed no potency in breast cancer. For instance, matrine could affect the cell mode of proliferation, programmed cell death, and autophagy in human liver cancer HepG2 cells [22]. Similarly, the same mechanisms were observed in the HT29 colon cancer cell line [23]. Oxymatrine could inhibit adenomas through inhibition of angiogenesis and up-regulating the expression of the NF-κB-mediated VEGF signaling pathway [24]. Sophoridine could also inhibit the growth and apoptosis of SW480 colorectal cancer cells [25]. Ironically, all these compounds, as well as the SF extracts, did not show any effects on the MDA-MB-231 cells. Thus, further experiments are necessary to clarify this phenomenon.

### 2.6. In-Vitro Acetylcholinesterase Inhibition Test

The AChE inhibition percentages at the concentration of 0.1 mg/mL of the SF total extract, the chloroform fraction, the pure compound oxysophocarpine, oxymatrine, matrine, and sophoridine was 69.22 ± 6.87%, 27.85 ± 7.91%, 21.14 ± 5.78%, 19.43 ± 7.64%, 17.00 ± 3.56%, and 25.90 ± 4.51%, respectively (Figure 4). Since only the SF total extract yielded adequate AChE inhibition efficacy, its IC50 was determined to be 0.1077 ± 0.0023 mg/mL. The obtained results were consistent with the previous studies. For instance, Jong Eun Park et al. (2022) reported that the SF 95%-alcohol extract, at the concentration of 20 µg/mL, exhibited an AChE inhibitory effect of 37–57%, better than the SF 70%-alcohol extract, the 50%-alcohol extract, and the water extract [26]. The lavandulyl groups presented in the SF extract compounds might play a key role in the extract AChE inhibitory activity [27]. This fact, together with the ability to inhibit the BACE1 enzyme [27], suggests the potential of SF extract in the prevention and/or treatment of dementia.

## 3. Materials and Methods

### 3.1. Materials

SF roots, grown in Dak Nong, Vietnam, from February 2017 to February 2019, were collected and analytically identified. The standard compounds matrine, oxymatrine, and sophoridine were imported from Chengdu Biopurify Phytochemicals Ltd., Sichuan, China. Ethanol (EtOH), methanol (MeOH), acetic acid (glacial) (AcOH), acetonitrile (ACN), chloroform, DMSO, and Tris hydrochloride buffer were purchased from Merck, Darmstadt, Germany. Acetylcholinesterase (AChE), 3-(4,5-dimethylthiazol-2-yl)-2,5-diphenyltetrazolium bromide) (MTT), and 1,1-diphenyl-2-picrylhydrazyl (DPPH) were purchased from Sigma-Aldrich, Burlington, MA, USA. Cells culture materials such as the DMEM medium, fetal bovine serum (PBS), L-glutamine, Penicillin–Streptomycin (PenStrep), and trypsin-EDTA were supplied by Gibco, Grand Island, NY, USA.

### 3.2. Plant Identifications

To identify the SF, the collected plant materials were first morphologically determined by a botanical expert based on the book “Plants of Vietnam”. Then, the DNA identification technique was employed to critically identify the SF. The plant voucher specimen was preserved at the University of Medicine and Pharmacy, Ho Chi Minh City, Vietnam.

For the plant DNA extraction, the JET Plant Genomic DNA Purification mini kit was utilized, with the standard procedure described by the manufacturer. Briefly, 20 mg of the dried SF was finely ground and extracted with Buffer AP1 and 4 µL RNase A. Then, the mixture was incubated at 65 °C for 15 min, and 130 µL of buffer P3 was subjected to the samples, followed by another 5-min incubation. Next, the mixture was centrifuged at 14,000 rpm for 2 min, and the filtrate was mixed with AW1 and AW2 buffer sequentially, followed by centrifugation at 14,000 rpm for 2 min. Finally, 100 µL of the AE buffer was incubated with the samples for 5 min at room temperature, and the mixture was centrifuged at 8000 rpm for 1 min to collect the DNA supernatant.

For the DNA amplification, 3 µL of the DNA supernatant (at the DNA amount of 0.1–1.0 µg) was mixed with two primers (1 µL, 0.25 nmol/µL) of ITS1 (5′-TCCGTAGGTGAACCTGCGG-3′) and ITS4 (5′-TCCTCCGCTTATTGATATGC-3′), and other ingredients instructed by the manufacturer. Then, the mixtures were subjected to polymerase chain reaction (PCR) amplifications, with 30 cycles at the temperature program of 94 °C for 30 s, 55 °C for the next 30 s, and 72 °C for 1 min. The amplified DNA was added to the agarose gel electrophoresis (1% agarose in TBE buffer). The gel was run in TBE buffer at 25 mV for 50 min. The DNA bands were elucidated under a UV lamp at a wavelength of 360 nm.

The gene sequencing was conducted at Nam Khoa Company LTD., Ho Chi Minh city, Vietnam, using the Sanger method on AIB’s 3130XL machine. The obtained DNA sequences were assembled using DNA STAR and compared with the DNA in the National Center for Biotechnology Information (NCBI) library.

### 3.3. Plant Extractions, Isolations, and Alkaloids Purifications

The simple maceration technique was employed to extract the SF. Briefly, 10 g of the dried SF fine powder was subjected to 200 mL of the extraction solvents for 2 h, repeated thrice, and the mixtures were filtered to obtain the SF total extract. The total extract was then dispersed in 50 mL water at pH 2, followed by partition thrice with 100 mL of ethyl acetate. The water portion was alkalized in NH_3_ to pH 10 and partitioned thrice with 100 mL of chloroform. Finally, the chloroform portion was condensed using a vacuum rotavapor to obtain the chloroform fraction. In this study, we investigated the effects of 9 different extraction solvents (EtOH, EtOH + 1% HCl, EtOH + 1% AcOH, MeOH, MeOH + 1% HCl, MeOH + 1% AcOH, H_2_O, H_2_O + 1% HCl, and H_2_O + 1% AcOH) on the extract yields.

The alkaloids in the chloroform fraction were further isolated and purified to identify their structures. To this end, the extract was first fractionally crystallized by adding an amount of ethylic ether 7–8 times the amount of CHCl_3_. Then, 4 g of the fractional crystals was dissolved in the solvent mixture of CHCl_3_-MeOH-NH_4_OH 25% (50:10:3, lower layer) and subjected to the column chromatography containing 40 g of medium-grain silica gel (Merck, particle size of 0.04–0.063 mm) as stationary phase. The mobile phase was the same solvent mixture used to dissolve the crystals. The fractions (50 mL) were collected, identified by thin-layer chromatography, and pooled together. The alkaloids in these fractions were then purified by reverse-phase preparative chromatography. For this, the alkaloids were reacted with maleic acid to form salt forms. Then, the products were subjected to an HPLC machine for separation and purification. All HPLC analyses were critically validated, in terms of specificity, accuracy, precision, linearity, range, and LOD/LOQ, following the ICH guidelines.

### 3.4. Isolated Compound Structure Elucidations

Prior to the elucidations of the SF isolated compounds, the purity of the isolates was re-confirmed with HPLC-PDA analysis (Shimadzu LC-2030C machine, Phenomenex Gemini NX C18 (250 × 21.2 mm, 5 μm) column, PDA detector at 210 nm, injection volume of 10 µL, flow rate of 1 mL/min, and a mobile phase of MeOH-trifluoroacetic acid 0.015% (10:90 *v*/*v*)). Then, the compound chemical structures were determined by spectroscopic techniques, including FTIR, MS, and NMR (^13^C-NMR and ^1^H-NMR). 

For the FTIR, the KBr pellet technique was employed on the Bruker Alpha T spectrophotometer (Bruker, Billerica, MA, USA). The machine was equipped with a DTGS detector, and the spectra were obtained at a co-addition of 256 interferograms collected at 4.0 cm^−1^ resolution, with a wavenumber range of 400–4000 cm^−1^.

For the MS, the compound mass was obtained using a Fourier-transform ion cyclotron resonance (FT-ICR) apparatus (Perkin-Elmer, Waltham, MA, USA) at room temperature utilizing the electrospray ionization (ESI) method.

For the NMR, the samples were measured in CDCl_3_ (δH 7.27 s; δC 77.0) performed on a Bruker Advance 500 MHz instrument. Signals were demonstrated on a ppm scale with tetramethylsilane (TMS) as an internal standard.

### 3.5. In-Vitro Antioxidant Test

The antioxidant activities of the extracts and the pure isolated compounds were determined by the DPPH and malondialdehyde (MDA) assays, following standardized protocols. Briefly, for the DPPH test, 100 µL of the test samples or the standard compound quercetin, at various concentrations, were mixed with 100 µL of 0.2 mM DPPH methanolic solution, followed by incubation in the dark at room temperature for 30 min. The mixture UV-Vis absorbance values at a wavelength of 517 nm were then measured. The antioxidant activity, in terms of the percentage of scavenged DPPH, was calculated by Equation (1).
(1)$$\mathrm{Antioxidant}\;\mathrm{activity}\;\left(\%\right)\;=\;\frac{\left(A_{0}-A_{1}\right)}{A_{0}}\times 100$$
where *A*_0_ and *A*_1_ are the absorbance value of the DPPH/MDA without and with the presence of the test/reference samples.

For the MDA assay, 50 µL of the test sample or the standard quercetin, at different concentrations, was mixed with 250 µL of 10% liver extract and filled up to 1 mL with 50 mM sodium phosphate buffer (pH 7.4). The mixtures were incubated at 37 °C for 60 min, and the reaction was halted with 500 µL of 10% trichloroacetic acid. Then, the solutions were centrifuged, and 500 µL of the supernatant was mixed with 500 µL of 0.8% thiobarbituric acid at 100 °C for 15 min. Finally, the UV-Vis absorbance of the samples was measured at the wavelength of 532 nm. The antioxidant activity was calculated by Equation (1).

### 3.6. In-Vitro Cytotoxicity Test

The cytotoxicity actions of the extracts and the pure isolated compounds were tested on the breast cancer cell lines MDA-MB-231 using the MTT assay. For this, cells were cultured in DMEM medium, supplemented with 10% FBS, 2 mM L-glutamine, 100 IU/mL penicillin, and 100 µM streptomycin, and incubated at 37 °C, 5% CO_2_ until 70–80% coverage. Cells were then transferred into 96-well culture plates with appropriate density. The plate was incubated for another 24 h at 37 °C, 5% CO_2_ for cell growth and adhesion. Then, the cells were treated with test samples prepared in culture media at different concentrations (100, 50, 25, and 12.5 µg/mL) for 72 h. The negative control (solvent) and the positive control (doxorubicin) were performed simultaneously. After 72 h, cells were washed with PBS solution thrice and incubated with 0.5 mg/mL MTT solution in serum-free culture medium at 37 °C, 5% CO_2_ for 3 h. Then, the formed formazan crystals were dissolved in the acidified isopropanol, and the solutions were UV-Vis spectroscopic measured at 570 nm using a microplate reader. The percentage of cell inhibition is calculated based on Equation (2).
(2)$$\%\;Inhibition\;=\;\left(1-\frac{Absorbance\;\left(sample\right)-Absorbance\;\left(blank\right)}{Absorbance\;\left(positive\;control\right)-Absorbance\;\left(blank\right)}\right)\times 100\%$$

### 3.7. In-Vitro Acetylcholinesterase Inhibition Test

For the acetylcholinesterase (AChE) inhibition test, the assays were performed on a 96-well plate at 0–4 °C. For this, 4 types of samples were prepared, including (1) the blank + enzyme (BE), (2) the blank (B), (3) the test sample + enzyme (CE), and the test sample xonly (C). For the BE sample, 180 µL of the 0.1 M Tris HCl pH 8.0 was mixed with 30 µL of 3.3 mM DTNB and 60 µL AChE 0.415 U/mL. For the B sample, 240 µL of the 0.1 M Tris HCl pH 8.0 was mixed with 30 µL of 3.3 mM DTNB. For the CE sample, 150 µL of the 0.1 ×M Tris HCl pH 8.0 was mixed with 30 µL of 3.3 mM DTNB, 60 µL AChE 0.415 U/mL, and 30 µL of the extracts/compounds. For the C sample, 210 µL of the 0.1 M Tris HCl pH 8.0 was mixed with 30 µL of 3.3 mM DTNB and 30 µL of the extracts/compounds. After the mixing, the wells were incubated for 15 min, followed by the addition of 30 µL ATCI solution, and incubated for another 1 min. Finally, the mixture absorbance values were determined at 415 nm using a Synergy Cooperative instrument (BioTek, Winooski, VT, USA). The percentage of AChE inhibition was calculated using Equation (3). The sample with the highest inhibitory efficacy was further determined by their 50%-inhibitory concentrations (IC50) using the curve-fitting method.
(3)$$\%\;AChE\;inhibition\;=\;\frac{Absorbance\;\left(BE\right)-Absorbance\;\left(B\right)}{Absorbance\;\left(CE\right)-Absorbance\;\left(C\right)}\;\times 100\%$$

### 3.8. Statistical Analysis

Each experiment was repeated at least three times, and the quantitative results were presented as mean ± standard deviation (Mean ± SD). Student’s *t*-tests and ANOVA were utilized to compare the values between samples, with *p* < 0.05 for significant comparisons.

## 4. Conclusions

This study investigated the extractions, isolations, and in-vitro therapeutic actions of the SF root extracts, as well as their purified alkaloid compounds. For this purpose, four pure compounds were successfully isolated and were identified to be oxysophocarpine, oxymatrine, matrine, and sophoridine. The extracts and the compounds did not possess significant antioxidant activity using the DPPH and MDA assays and cytotoxicity action using the MTT assay on the MDA-MB-231 breast cancer cell line. On the other hand, the SF total extract yielded a moderate AChE inhibition effect with an IC50 of 0.1077 ± 0.0023 mg/mL. In conclusion, SF could be further investigated to be a potential medicinal source for the treatment of dementia.

## Figures and Tables

**Figure 1 pharmaceuticals-15-01384-f001:**
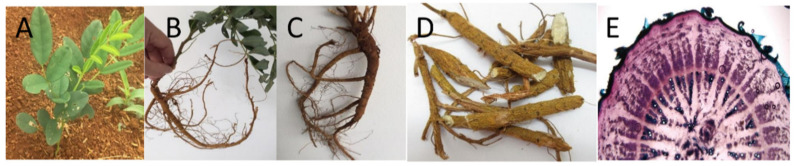
The *Sophora flavescens* Ait. plant. (**A**) Fresh plant; (**B**,**C**) fresh root; (**D**) dried root; (**E**) root cross-section.

**Figure 2 pharmaceuticals-15-01384-f002:**
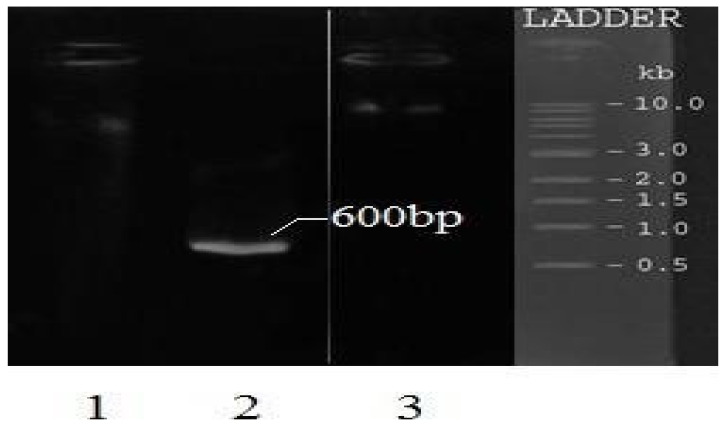
The *Sophora flavescens* Ait. plant DNA gel electrophoresis. (1) The blank sample; (2) the PCR-amplified DNA sample; (3) the total DNA sample.

**Figure 3 pharmaceuticals-15-01384-f003:**
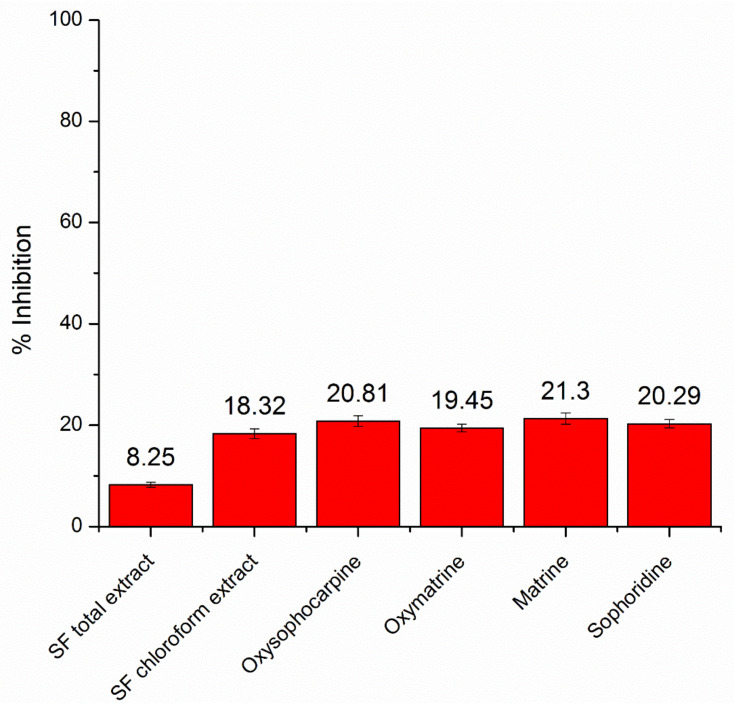
The in-vitro cytotoxicity, in terms of percentages of inhibition, at the sample concentration of 100 µg/mL of the *Sophora flavescens* Ait. (SF) total extract, the chloroform fraction, and the 4 isolated pure compounds (*n* = 3).

**Figure 4 pharmaceuticals-15-01384-f004:**
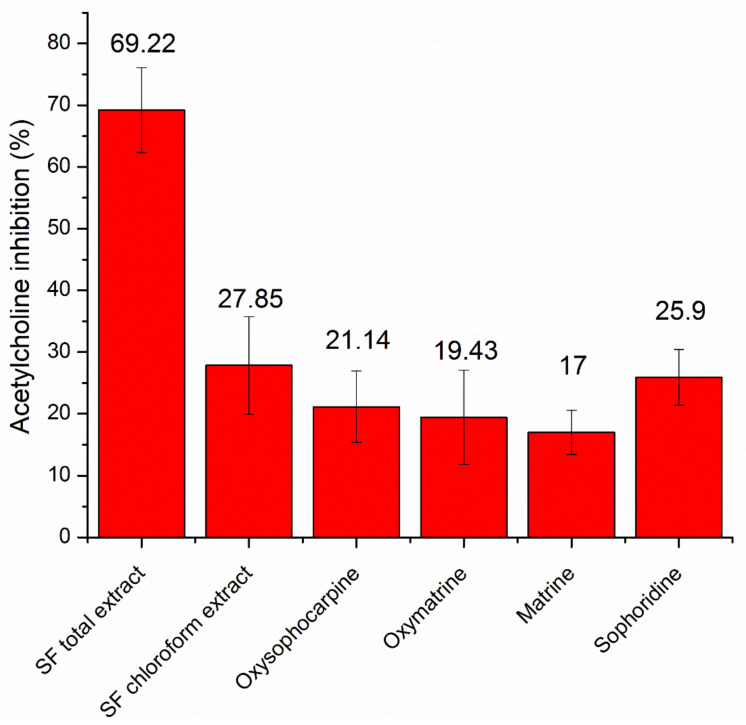
The acetylcholinesterase inhibitory actions (%), at the sample concentration of 0.1 mg/mL of the *Sophora flavescens* Ait. (SF) extract, the chloroform fraction, and the 4 isolated pure compounds (*n* = 3).

**Table 1 pharmaceuticals-15-01384-t001:** Effects of different extraction solvents (water, ethanol (EtOH), and methanol (MeOH)), with or without the acidic reagents of HCl or acetic acid (AcOH), on the *Sophora flavescens* Ait. total extract and chloroform fraction yields (*n* = 3). ^a, b, c^ different letters denote significant differences between values in the same column (*p* < 0.05).

Extraction Solvent	Total Extract (g)	Chloroform Fraction (g)
EtOH	2.27 ± 0.30 ^a^	0.20 ± 0.02 ^a^
EtOH + 1% HCl	4.32 ± 0.35 ^b^	0.29 ± 0.03 ^b^
EtOH + 1% AcOH	2.55 ± 0.28 ^a^	0.27 ± 0.03 ^b^
MeOH	2.20 ± 0.27 ^a^	0.16 ± 0.02 ^a^
MeOH + 1% HCl	3.03 ± 0.24 ^c^	0.19 ± 0.02 ^a^
MeOH + 1% AcOH	2.57 ± 0.22 ^a^	0.20 ± 0.02 ^a^
H_2_O	2.31 ± 0.21 ^a^	0.20 ± 0.03 ^a^
H_2_O + 1% HCl	4.55 ± 0.39 ^b^	0.22 ± 0.03 ^a^
H_2_O + 1% AcOH	2.49 ± 0.25 ^a^	0.20 ± 0.02 ^a^

**Table 2 pharmaceuticals-15-01384-t002:** Chemical properties, in terms of the mass (measured by mass spectrometry-MS) and the infrared spectra (FTIR) of the 4 isolated compounds (A1–A4) from *Sophora flavescens* Ait.

	A1 (Oxysophocarpine)	A2 (Oxymatrine)
Chemical formula	C_15_H_22_N_2_O_2_	C_15_H_24_N_2_O_2_
Chemical structure	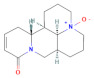	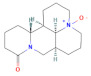
Mass (MS)	MS: *m*/*z* = 263.1711 [M + H]^+^ Theory: 263.1747	MS: *m*/*z* = 265.1893 [M + H]^+^ Theory: 265.18986
FTIR	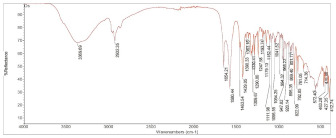	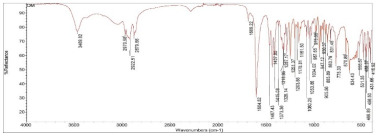
	**A3 (Matrine)**	**A4 (Sophoridine)**
Chemical formula	C_15_H_24_N_2_O	C_15_H_24_N_2_O
Chemical structure	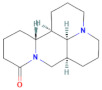	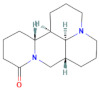
Mass (MS)	MS: *m*/*z* = 249.19554 [M + H]^+^ Theory: 249.1955	MS: *m*/*z* = 249.19488 [M + H]^+^ Theory: 249.1950
FTIR	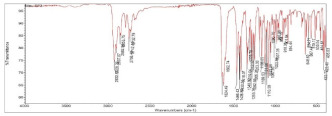	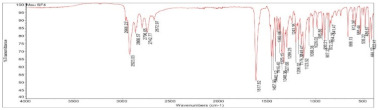

**Table 3 pharmaceuticals-15-01384-t003:** Nuclear magnetic resonance (NMR) spectral peaks (carbon and hydrogen) of the 4 isolated compounds (A1–A4) from *Sophora flavescens* Ait., identified as oxysophocarpine, oxymatrine, matrine, and sophoridine, and the corresponding references.

C	A1 (Oxysophocarpine) [18]			A2 (Oxymatrine) [19]			A3 (Matrine) [19]			A4 (Sophoridine) [20]		
	*δ*_C_ (A1)	*δ*_H_ (A1)		*δ*_C_ (A2)	*δ*_H_ (A2)		*δ*_C_ (A3)	*δ*_H_ (A3)		*δ*_C_ (A4)	*δ*_H_ (A4)	
2	69.0	3.0–3.2	3.0–3.2	69.6	3.12	3.19	58.4	2.83	-	57.5	2.08	2.79
3	17.2	2.6–2.8	1.5–1.9	17.3	2.75	1.56	22.2	-	-	23.6	1.96	-
4	26.2	1.5–1.9	1.5–1.9	26.2	1.7	1.8	28.1	-	-	28.7	1.87	-
5	33.6	1.8–2.1	-	34.6	1.86	-	36.9	-	-	31.5	1.96	-
6	67.1	3.0–3.2	-	67.3	3.06	-	65.1	-	-	63.6	2.27	-
7	40.7	1.8–2.1	-	42.8	1.58	-	42.9	-	-	41.7	-	-
8	24.9	1.8–2.1	1.5–1.9	24.8	1.56	2.05	27.3	-	-	22.9	-	-
9	17.2	2.6–2.8	1.5–1.9	17.3	2.67	1.54	21.7	-	-	22.9	-	-
10	69.3	3.0–3.2	3.0–3.2	69.2	3.09	3.17	58.3	2.83	-	50.7	2.08	2.79
11	51.6	5.09	-	53.0	5.09	-	54.8	3.83	-	56.7	3.4–3.2	-
12	28.9	2.6–2.8	1.8–2.1	28.6	1.26	2.2	28.8	-	-	30.7	1.87	-
13	137.1	6.46	-	18.7	1.69	1.8	19.6	-	-	19.5	-	-
14	125.0	5.91	-	33.0	2.26	2.45	33.4	-	-	33.0	2.27	-
15	166.4	-	-	170.2	-	-	172.1	-	-	172.6	-	-
17	42.6	4.17	4.08	41.8	4.17	4.41	44.7	4.30	3.07	48.9	3.4–3.2	-

**Table 4 pharmaceuticals-15-01384-t004:** In-vitro antioxidant activities, using the DPPH and malondialdehyde (MDA) assays of the *Sophora flavescens* Ait. (SF) total extract, the SF chloroform fraction, and the pure isolated compounds (oxysophocarpine, oxymatrine, matrine, and sophoridine) (*n* = 3).

Sample	DPPH Test (%)	MDA Test (%)
SF total extract, 1000 µg/mL	6.35 ± 0.32	3.38 ± 0.11
SF chloroform fraction, 1000 µg/mL	23.19 ± 1.86	48.88 ± 0.63
Oxysophocarpine, 1000 µg/mL	5.26 ± 2.34	5.73 ± 1.68
Oxymatrine, 1000 µg/mL	5.46 ± 3.13	10.61 ± 4.94
Matrine, 1000 µg/mL	8.15 ± 3.78	12.26 ± 4.50
Sophoridine, 1000 µg/mL	22.02 ± 2.80	18.57 ± 5.19

## Data Availability

Data is contained within the article and Appendix A.

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
