# Peer review of "Determination of Antioxidant, Cytotoxicity, and Acetylcholinesterase Inhibitory Activities of Alkaloids Isolated from Sophora flavescens Ait. Grown in Dak Nong, Vietnam"

_pharmaceuticals, 2022, doi:10.3390/ph15111384_

Round 1

Reviewer 1 Report

The manuscript written by Thang et al is of interest but should be improved prior publication. The cellular effects of the compounds should be shown (i.e., the MTT assay). Moreover, microphotographs of treated cells should be provided to estimate judge the effects or absence thereof.

Author Response

1. The manuscript written by Thang et al is of interest but should be improved prior publication. The cellular effects of the compounds should be shown (i.e., the MTT assay).

Thanks for your suggestion. We have added these information in the manuscript.

2. Moreover, microphotographs of treated cells should be provided to estimate judge the effects or absence thereof.

Thank you. To be honest, we did not take these cell photographs because they did not look significantly different when comparing between the untreated and treated cells. Moreover, due to the fact that our extracts/compounds did not possess moderate cytotoxic actions on this cell line, thus, in our humble opinion, we think that it is unnecessary to add the images of the treated cells, as they are no different than the untreated cells. We sincerely apologize for not following your suggestion.

Reviewer 2 Report

rows 92-94 - the phrase is not understandable, please rephrase

the results of in vitro cytotoxicity tests must be supported by proofs (figures, tables aso)

the antioxidant activity of various fractions should be also presented as data in a table or as graphic.

the results of the acetylcholinesterase inhibition test must be presented properly as table or graphic, not as a figures row.

the paper must be completed with all the data resulted from your research presented in a proper and convincing way

Author Response

1. rows 92-94 - the phrase is not understandable, please rephrase

Thank you. We have adjusted the phrase and sentence to be easier to understand.

2. the results of in vitro cytotoxicity tests must be supported by proofs (figures, tables aso)

Thanks for your comment. We have added more information regarding this issue in the in-vitro cytotoxicity section.

3. the antioxidant activity of various fractions should be also presented as data in a table or as graphic.

Thank you for your suggestion. More data and a Table (Table 4) were added on this issue.

4. the results of the acetylcholinesterase inhibition test must be presented properly as table or graphic, not as a figures row.

Thank you! We have presented these data in a new figure.

5. the paper must be completed with all the data resulted from your research presented in a proper and convincing way

Thanks very much. We have critically revised the article, with more appropriate data, figures, tables, and texts.

Reviewer 3 Report

The manuscript by Pham is well-organized. However, I think the topic of this paper doesn't fit the scope of this journal. I suggest the author find a more suitable journal to publish their manuscript.

Author Response

The manuscript by Pham is well-organized. However, I think the topic of this paper doesn't fit the scope of this journal. I suggest the author find a more suitable journal to publish their manuscript.

Thanks so much for your comment. The “Aims and Scope” section of the Pharmaceuticals journal, published in the website, states that “The multidisciplinary journal welcomes manuscripts covering a wide range of aspects involved in drug discovery and development. The following topics are considered: ... Biomolecules, natural products, phages, and cells as therapeutic tools: peptides, aptamers, glycans, antibodies, extracts, bacteriophages, and stem cells ...”. Thus, we humbly think that our article, which focuses on the natural products, extracts, and their corresponding biological actions, is within the scope of the journal.

Reviewer 4 Report

Please find my comments in the attached file

Author Response

1. The title: Please change “anticancer” into “cytotoxicity”

Cytotoxicity is an in vitro test on the cell line, but anticancer or antitumor is applied when we talk about in vivo study.

Please apply the same thing through the whole text.

Thanks for your comment. We have adjusted this words throughout the whole manuscript.

2. Abstract: Line 30-33: Please rephrase “In summary, the SF extract demonstrated potential effects on the acetylcholinesterase-related diseases such as dementia, and could be further researched to become a pharmaceutical product for these diseases treatments.” “In summary, the SF extract demonstrated potential effects as anti-acetylcholinesterase, and could be further
researched to become a pharmaceutical product for diseases related to acetylcholine deficiency such as dementia.”

Your results don’t support this claim. “the SF extract demonstrated potential effects on the acetylcholinesterase-related diseases such as dementia”

Thank you. We have rephrased these sentences to avoid further misunderstanding.

3. Introduction: As shown in literature, the plant has been researched; many alkaloids were isolated and the same type of biological were reported for the plant extract, fraction and pure compounds.
https://doi.org/10.1248/bpb.29.1911; https://doi.org/10.1016/j.phymed.2019.152852;
https://doi.org/10.1142/S0192415X10007944
Therefore, you should introduce more exciting introduction regarding your aim from the research, and why you choose to carry out these biological studies specifically

Thanks for your suggestion. Although the fact that SF has been previously investigated, in-detailed research on SF cultivated in Vietnam are limited. Moreover, to the best of our knowledge, the anti-acetylcholinesterase effects of the SF alkaloid-rich extract, as well as its pure isolated alkaloids, have not yet been investigated. Thus, we explored these issues in this study. We have re-written the Introduction to add these information, and to thoroughly clarify the research aims. Thanks again.

4. Plant identifications: The flower features are the main route to identify the plant. Why there is no description for the plant flower, please add.

Thanks for your comment. You are correct! The flower features are one of the main route to identify the plant. However, in our humble opinion, we think that the DNA sequencing technique is a more powerful and accurate method to identify a plant. Thus, we focused on this method as a main route, without considering much on the flower characteristics, and we did not have these data, to be honest. Therefore, we sincerely apologize that we could not add these information in the manuscript. We hope you would understand.

5. Line 79: cross-section had protrusions resembling air vessels

Please provide figure for the histological study even as supplementary data.

Thanks for your comment, we have added this figure in Figure 1.

6. Line 80: Please add reference

Thank you. We have added the reference for this sentence

7. Figure 2: Why the PCR-amplified DNA sample and the total DNA sample are not shown the same gel?

Thanks for your question. We sincerely apologize in case we misunderstand your point. The PCR-amplified DNA is the DNA sample that was augmented/repeated millions/billions times by the PCR technique; and the total DNA is the DNA sample that was extracted directly from the plant, without any amplification. Thus, the main DNA band of the plant might not appear in the total DNA sample (due to undetectable amount), yet could appear clearly in the PCR-amplified DNA sample (due to the high amount of replications). We hope that we have answered your question correctly.

8. Table 1. Please change chloroform extract into chloroform fraction because it was not taken directly from the plant. Please make the same change throughout the manuscript.

Thanks for your suggestion. We have adjusted these words in the entire manuscript.

9. Figures 3-5 are better shown as supplementary data.

Thanks for your comment. We have moved these three figures to the supplementary file, and cited them in-text accordingly.

10. Line 133. Then, these 04 compounds were fully elucidated their structures. We use structure elucidation when the compound is new, pleas change “elucidated” into “identified”

Thank you, we have adjusted this word throughout the manuscript.

11. Table 2. Please enhance resolution of the structure.

Why are some atoms coloured? Is that have special mean?

Thank you. We have enhanced the chemical structure resolutions of compounds A1-A4. The atoms were colored to notify their chemical configuration (i.e., R/S isomers) and/or their element that different than the common carbon atoms.

12. Line 138: A1, A2, A3, and A4 .→ change into A1‒A4. Please apply the same whenever repeated.

Thanks for your comment. We have changed it accordingly.

13. Please add labels for the IR bands that recognize the different stretching and bending stretches on the IR spectra.

Thanks for your suggestion. We humbly think that most of the peaks of the IR spectra of these compounds (A1-A4) located in the fingerprint areas (600-1400 cm-1). Thus, it would be abundant to add the stretching and bending signals on the IR spectra, which could make them more complicated. Therefore, we sincerely apologize for not following your suggestions in this issue.

14. Table 3. Please change δH and δC → δH and δC. Please apply the same whenever repeated.

Since the compounds are known, please remove values obtained from reference; you just refer to the references that you considered to confirm identity of your compounds.

Thank you. We have made these changes accordingly, following your suggestion. The Table 3 is now adjusted to be better and more comprehensive. Thanks again!

15. Line 142. In-vitro antioxidant test.

Thanks, we have adjusted it.

16. Line 146: This indicates that the SF roots might not be a potential antioxidant source.

However, previous study detected antioxidant activity for the SF extract and fraction.

https://doi.org/10.1248/bpb.29.1911

How to explain the difference in your results from the previous study?

The same question for the cytotoxicity results, as previous study report promising cytotoxic activity against the same cell line and other. https://doi.org/10.1016/j.phymed.2019.152852

Thanks for your question. For the antioxidant activity, indeed, the previous study, as you mentioned, has demonstrated SF antioxidant activity using the DPPH assay. Compared to our results, this different result could be attributed to the different plant chemical compositions in their study and our study. Specifically, the SF from their study was collected in China, whereas in the SF in our work was collected in Vietnam. Since different geographical locations, as well as the soil nutrition, and other relevant factors, could significantly affect the SF chemical content, its antioxidant activity could also be affected. We have added these discussions in the manuscript, where relevant. Thank you!

For the cytotoxicity results, we have checked again your reference, as well as other previous works, and we found out that none of them reported the cytotoxicity action of the SF extracts and its 04 alkaloids (oxysophocarpine, oxymatrine, matrine, and sophoridine) on the MDA-MB-231 cell line. For your reference, they tested with the same cell line, but with different compound, the Sophoraflavanone G.

Round 2

Reviewer 2 Report

the results of cytotoxicity assay should be better presented 

Author Response

The results of cytotoxicity assay should be better presented

Thanks for your suggestion. We have added a Figure (Figure 3) regarding these information in the manuscript. The corresponding in-text citation was also adjusted.

Reviewer 3 Report

The author has made substantial revisions on this manuscript, the current form of manuscript is acceptable for publication.

Author Response

The author has made substantial revisions on this manuscript, the current form of manuscript is acceptable for publication.

Thanks for your comment and all your hard works to help us improving our article.